# microRNA–mRNA Networks Linked to Inflammation and Immune System Regulation in Inflammatory Bowel Disease

**DOI:** 10.3390/biomedicines12020422

**Published:** 2024-02-12

**Authors:** Carina de F. de Síbia, Ana E. V. Quaglio, Ellen C. S. de Oliveira, Jéssica N. Pereira, Jovita R. Ariede, Rainer M. L. Lapa, Fábio E. Severino, Patricia P. Reis, Lígia Y. Sassaki, Rogerio Saad-Hossne

**Affiliations:** 1Department of Surgery and Orthopedics, Botucatu Medical School, São Paulo State University (UNESP), Botucatu 18618-687, SP, Brazil; carinasibia@gmail.com (C.d.F.d.S.); jovitaariede@gmail.com (J.R.A.); fabio.bjj@gmail.com (F.E.S.); patricia.reis@unesp.br (P.P.R.); 2Laboratory of Phytomedicines, Pharmacology and Biotechnology (PhytoPharmaTec), Department of Biophysics and Pharmacology, Institute of Biosciences, São Paulo State University (UNESP), Botucatu 18607-440, SP, Brazil; anaequaglio@hotmail.com; 3Department of Internal Medicine, Botucatu Medical School, São Paulo State University (UNESP), Botucatu 18618-687, SP, Brazil; jessica.n.pereira@unesp.br (J.N.P.); ligia.sassaki@unesp.br (L.Y.S.); 4Experimental Research Unity (UNIPEX), Botucatu Medical School, São Paulo State University (UNESP), Botucatu 18618-687, SP, Brazil; 5Facultad de Ingeniería Zootecnista, Agronegocios y Biotecnología, Instituto de Investigación en Ganadería y Biotecnología, Universidad Nacional Toribio Rodríguez de Mendoza de Amazonas, Chachapoyas 01001, Peru; rainer.lopez@untrm.edu.pe; 6Facultad de Ciencias de la Salud, Instituto de Investigación de Salud Integral Intercultural, Universidad Nacional Toribio Rodríguez de Mendoza de Amazonas, Chachapoyas 01001, Peru

**Keywords:** inflammatory bowel disease, Crohn’s disease, ulcerative colitis, microRNA, meta-analysis

## Abstract

The molecular processes linked to the development and progression of Crohn’s disease (CD) and ulcerative colitis (UC) are not completely understood. MicroRNAs (miRNAs) regulate gene expression and are indicated as diagnostic, prognostic, and predictive biomarkers in chronic degenerative diseases. Our objectives included the identification of global miRNA expression in CD and UC, as well as miRNA target genes, miRNA–mRNA interaction networks, and biological functions associated with these different forms of inflammatory bowel disease (IBD). Methods: By performing a comprehensive meta-analysis, we integrated miRNA expression data from nine studies in IBD. We obtained detailed information on significantly deregulated miRNAs (fold change, FC ≥ 2 and *p* < 0.05), sample type and number, and platform applied for analysis in the training and validation sets. Further bioinformatic analyses were performed to identify miRNA target genes, by using the microRNA Data Integration Portal tool. We also sought to identify statistically enriched pathways of genes regulated by miRNAs using ToppGene Suite. Additional analyses were performed to filter for genes expressed in intestinal tissue using the European Bioinformatics Institute (EBI) database. Results: Our findings showed the upregulation of 15 miRNAs in CD and 33 in UC. Conversely, six miRNAs were downregulated in CD, while seven were downregulated in UC. These results indicate a greater deregulation of miRNAs in UC compared to CD. Of note, miRNA target genes were enriched for immune system regulation pathways. Among significantly deregulated miRNAs with a higher number of miRNA–target gene interactions, we identified miR-199a-5p and miR-362-3p altered in CD, while among UC case patients, miRNA–target gene interactions were higher for miR-155-5p. Conclusions: The identified miRNAs play roles in regulating genes associated with immune system regulation and inflammation in IBD. Such miRNAs and their target genes have the potential to serve as clinically relevant biomarkers. These findings hold promise for enhancing the accuracy of diagnoses and facilitating the development of personalized treatment strategies for individuals with various forms of IBD.

## 1. Introduction

Crohn’s disease (CD) and ulcerative colitis (UC) are the main types of inflammatory bowel disease (IBD) that affect the gastrointestinal tract, including the small intestine and colon [1]. The origin of IBD is primarily ascribed to an abnormal immune reaction to the bacterial microbiome in the intestine, accompanied by alterations in the integrity of the intestinal mucosal barrier [1].

A comprehensive understanding of the etiology of IBD will likely benefit from a thorough exploration of gene regulation mechanisms that may impact disease development and progression. Currently, significant attention is directed toward elucidating these intricate processes, with interest in non-coding RNAs, including microRNAs (miRNAs), as pivotal players linked to IBD pathogenesis. miRNAs are small RNAs that act in the post-transcriptional regulation of gene expression [2]. miRNAs have been extensively associated with various processes, including cell proliferation, apoptosis, and differentiation, and exhibit distinct expression patterns in developing tissues [3,4]. miRNAs play roles in the development and progression of health conditions, exhibiting specific expression patterns in both tissues and particular diseases. Therefore, changes in miRNA expression have been indicated as clinically useful biomarkers to aid in the diagnosis, prognosis, and prediction of treatment response [4].

Inflammatory factors may impact miRNA expression levels prior to the clinical onset of IBD symptoms. miRNAs are promising biomarker candidates associated with IBD, as they may exhibit expression changes based on the specific type and stage of disease [5]. Studies have consistently demonstrated miRNA alterations in IBD, underscoring their sensitivity and specificity as dependable biomarkers capable of distinguishing between different disease subtypes [5,6,7].

To the best of our knowledge, initial studies of altered miRNA expression in IBD were reported by Wu et al. [3]. In their study, patients with active UC exhibited elevated levels of eight miRNAs (miR-16, miR-21, miR-23a, miR-24, miR-29a, miR-126, miR-195, and Let-7f) and reduced levels of three miRNAs (miR-192, miR-375, and miR-422b) in the intestinal mucosa compared to healthy individuals [3]. Furthermore, Coskun et al. [8] reported the deregulation of miR-20b, miR-98, miR-125b-1, and let-7e in patients with active UC, when compared to individuals with CD and healthy volunteers. 

In addition to solid tissues, miRNAs can also be extracted from cells and body fluids, such as serum and plasma. A study that analyzed miRNA expression in extracellular vesicles, circulating immune cells, and blood platelets, reported 31 deregulated miRNAs in UC patients compared to healthy individuals, with a high accuracy rate of 96.2% specificity and 89.5% sensibility [2], indicating their potential application as non-invasive UC diagnostic biomarkers.

While the above-mentioned studies suggest the involvement of miRNAs in IBD, most of them have not employed comprehensive global miRNA expression analyses. Instead, they have examined candidate miRNAs or small miRNA panels, which represent only approximately 10% of the total known miRNAs in the human genome. Additionally, many studies lack robust data validation and do not establish correlations between miRNA–mRNA targets and altered biological functions observed in IBD. 

The identification of miRNA target genes is a valuable strategy in elucidating the functional significance of miRNA–gene networks implicated in IBD pathophysiology [9]. We performed a correlation analysis integrating miRNA expression with validated target genes that are found expressed in bowel tissue. Our goal was to gain valuable insights into the molecular pathways underpinning the pathophysiology of Crohn’s disease and ulcerative colitis, aiming to contribute to the search for future forms of diagnosis and/or treatment of IBD.

## 2. Material and Methods

### 2.1. Search Strategy

Our search strategy was performed using publicly available data on PubMed (http://www.ncbi.nlm.nih.gov/pubmed (accessed on 10 October 2023)). Data were retrieved based on the keywords: “microRNA AND inflammatory bowel disease (mesh terms)”, “microRNA AND Crohn’s disease (mesh terms)”, and “microRNA AND ulcerative colitis (mesh terms)”. We conducted an extensive literature search employing specific filters. Articles within the past 15 years, focusing on human disease, published in English, and available as free full text were considered. Subsequently, inclusion criteria were applied, encompassing studies analyzing global expression of miRNAs in inflammatory bowel disease, including Crohn’s disease and ulcerative colitis. These studies involved various sample types, such as intestinal mucosa tissues, plasma, or serum, and were restricted to diseases affecting human adults. Additionally, inclusion criteria required studies to include control samples from healthy individuals, incorporate data validation methods, and provide data either within the manuscript text, Appendix A, or through downloadable raw data. Exclusion criteria encompassed studies exclusively utilizing experimental models (in vitro or in vivo), literature reviews, or case reports.

### 2.2. Data Extraction

The meta-analysis study adhered to the guidelines outlined in the PRISMA Statement, as illustrated in Figure 1. We incorporated nine pertinent studies [2,3,8,10,11,12,13,14,15] (refer to Table 1), which were selected based on predefined inclusion and exclusion criteria. The gathered data encompassed information including the list of significantly deregulated miRNAs (with fold change ≥ 2 and *p*-value ≤ 0.05), the direction of deregulation (over- or under-expressed), the type of samples analyzed (plasma, serum, and/or colonic tissue obtained via biopsy or surgery), the quantity of samples from patients with Crohn’s disease and/or ulcerative colitis in both test and validation sets, the platform employed for miRNA expression analysis and validation, as well as details such as the first author’s name and date of publication. A comprehensive presentation of the collected data is provided in Table 1.

### 2.3. miRNA Target Prediction and Pathway Identification

Significantly deregulated miRNAs (FC ≥ 2 and *p* ≤ 0.05) were searched to identify their predicted mRNA targets. Additionally, we considered miRNA target genes that are expressed in colonic tissue or plasma. miRNA target prediction was carried out in the MicroRNA Data Integration Portal (http://ophid.utoronto.ca/mirDIP/ (accessed on 10 October 2023)) [16], a bioinformatic tool that allows consolidating sequence information and target prediction data from diverse sources, thereby enhancing the robustness of our investigative framework. To uphold stringency in target selection, we considered miRNA target genes with “very high” (top 1%) score results for interaction probability. 

Subsequently, we identified enriched pathways of miRNA target genes using the ToppGene Suite tool (https://toppgene.cchmc.org/ (accessed on 10 October 2023)) [17]. Furthermore, we conducted searches for the identified miRNA target genes in the European Bioinformatics Institute (EBI) database (https://www.ebi.ac.uk/gxa/home (accessed on 10 October 2023)). This resource provided comprehensive insights into gene expression patterns, allowing us to validate the expression of genes targeted by miRNAs and that are expressed in intestinal tissue [18]. This step was crucial as we meticulously filtered our data to focus on genes that were considered relevant to IBD pathogenesis. Finally, to visualize miRNAs specific to each condition (Crohn’s disease or ulcerative colitis) and those common to both forms of IBD, we constructed a Venn diagram using an interactive Venn tool.

## 3. Results

### 3.1. Differential miRNA Expression Profiles Characteristic of Crohn’s Disease vs. Ulcerative Colitis

Our focus was to discern significantly deregulated miRNAs, considering those with a fold expression change ≥ 2 (under- or over-expressed) and *p* ≤ 0.05, compared to controls. To comprehensively analyze a diverse array of literature research findings while upholding our predetermined inclusion criteria, we systematically reviewed articles on global miRNA expression alterations within plasma and intestinal tissues of individuals diagnosed with CD and/or UC. By employing a stratified approach based on disease classification, we effectively segregated miRNA expression data into two distinct categories: those linked to CD and those associated with UC. This analysis facilitated the identification of miRNAs with disease-specific deregulation, as well as those manifesting shared deregulation across both conditions. 

In our analysis of CD, we identified 25 deregulated miRNAs, comprising 15 over-expressed and 6 under-expressed miRNAs, and 4 miRNAs that were expressed in opposite ways depending on the location analyzed, as summarized in Table 2. Conversely, UC samples exhibited a more extensive miRNA deregulation profile, with 44 miRNAs displaying altered expression, surpassing the number observed in CD. Among these, 33 miRNAs were upregulated, 7 miRNAs were downregulated, and 4 were expressed in opposite ways depending on the location analyzed, as detailed in Table 3. Visual representations of the overlapping and unique sets of over-expressed and under-expressed miRNAs in these diseases can be observed in Figure 2.

### 3.2. miRNA–mRNA Networks and Biological Processes in Crohn’s Disease and Ulcerative Colitis

Identified miRNA–target gene interactions indicated the involvement of a large number of genes with biological functions potentially associated with IBD development and progression (the detailed list of miRNAs and their predicted targets is described in Appendix A).

In Crohn’s disease, the 15 over-expressed miRNAs were predicted to regulate 6906 genes with 14,515 interactions among these genes (Appendix A), and the 6 under-expressed miRNAs were predicted to regulate 7767 target genes with 15,880 interactions (Appendix A).

In ulcerative colitis, the 33 over-expressed miRNAs were predicted to regulate 10,710 genes with 45,552 interactions among genes (Appendix A), and the 7 under-expressed miRNAs were predicted to regulate 7171 genes having 18,782 interactions (Appendix A).

Computational analysis of genes for Gene Ontology annotation and pathway prediction revealed that these genes play crucial biological roles in pathways primarily associated with inflammation, immune system regulation, and cell trafficking interference, among others. In the context of CD, genes predicted to be regulated by over-expressed miRNAs were notably involved in enriched pathways related to the adaptive immune system, cytokine signaling in the immune system, interleukin signaling, inflammation mediated by chemokines and cytokines, as well as epidermal growth factor (EGF) and nerve growth factor (NGF) signaling pathways (Appendix A).

Regarding under-expressed miRNAs in CD, target genes were also found to be associated with inflammatory pathways, but among the most significantly altered pathways (top 10) were axon guidance, pathways in cancer, membrane trafficking, and signaling by NGF, among others (Appendix A). Interestingly, pathways associated with target genes of over-expressed miRNAs in ulcerative colitis (UC) mirrored those of genes regulated by over-expressed miRNAs in CD. These pathways included adaptive immune system responses, cytokine signaling in the immune system, interleukin signaling, inflammation mediated by chemokines and cytokines, as well as axon guidance (Appendix A).

When examining the pathways of target genes regulated by under-expressed miRNAs in UC, we identified that among the top 10 altered pathways were axon guidance, pathways in cancer, membrane trafficking, signaling by NGF, and signaling regulating pluripotency of stem cells (Appendix A).

## 4. Discussion

Herein, we elucidated miRNAs that are consistently deregulated in two prominent inflammatory bowel diseases, CD and UC. miRNAs are known regulatory molecules that influence cellular processes such as maturation, differentiation, apoptosis, and migration, playing a profound role in gene expression regulation [19]. Our findings unveiled distinct patterns of deregulated miRNAs in these diseases, with both over-expressed and under-expressed miRNAs contributing to their molecular landscape. In addition to identifying miRNA expression changes, the study of miRNA–target gene interactions is pivotal to clarify the molecular pathogenesis of IBD and to further explore novel avenues for more efficient treatment of patients suffering from the different forms of IBD.

A molecular diagnostic panel consisting of specific miRNAs may hold the potential to distinguish between CD and UC [13]. Furthermore, the effects of miRNA target gene deregulation impacting biological functions and pathogenesis of IBD remain poorly understood. We showed that a number of predicted miRNA target genes control inflammation and immune responses, among other pathways potentially associated with the pathophysiology of IBD. It is noteworthy that our target gene prediction, conducted through the miRNA Data Integration Portal (miRDIP) [16], a comprehensive tool amalgamating diverse data sources, was further bolstered by a stringent filtering approach to identify genes with precision and high probability of interaction. Additionally, leveraging tissue-specificity data from the European Bioinformatics Institute database, with a specific focus on validated genes expressed in intestinal tissue, contributed significantly to the accuracy and relevance of our target gene identification.

We showed that the predicted target genes of over-expressed miRNAs in CD and UC exhibited a noteworthy convergence towards common pathways, prominently featuring adaptive immune system responses, cytokine signaling within the immune system, interleukin signaling, and inflammation mediated by chemokines and cytokines. Additionally, these genes also participated in pathways involving EGF and NGF signaling. Similarly, under-expressed miRNAs in both CD and UC were associated with analogous pathways, mainly axon guidance, pathways in cancer, membrane trafficking, and signaling by NGF. These findings suggest that there are shared molecular mechanisms, at least in part, which underpin the pathogenesis of CD and UC, shedding light on potential therapeutic targets and avenues for further research in these complex diseases. 

In IBD, deregulated genes overstimulate the immune system causing an intense, permanent inflammatory response [20]. The onset of chronic inflammation is primarily attributed to deregulation in the expression of cytokines with pro- and anti-inflammatory properties [21].

Integrating genetic factors with environmental stimuli initiates the inflammatory process, marked by the recruitment of defense cells and the secretion of cytokines, including tumor necrosis factor (TNF-α) and interleukins (IL-4, IL-6, IL-12, IL-17, IL-23) [22]. Deregulation of miRNAs contributes to altered production of cytokines, thereby modulating their action [23]. For example, miR-21 targets the TNF-α pathway, enhancing inflammation; the knockout of this regulator can improve patients’ conditions [24]. miR-320 inhibits inflammation by targeting nucleotide-binding oligomerization domain 2 (NOD2); when downregulated, it can contribute to the onset of inflammation [25]. The clinical importance of elucidating miRNA–target gene interaction is the possibility to apply this knowledge in developing novel target-driven therapies focusing on key molecules, aiming to reduce relapses and improve therapeutic responses.

Bai et al. [26] reported molecular alterations associated with disease progression in CD, UC, and colorectal carcinoma. They showed a set of deregulated miRNAs (miR-125b, miR-335, and miR-155) associated with metabolic pathways. Also, Toll-like receptor (TLR) signaling genes were predicted as regulated by miR-124, miR-146a, and miR-221/222. Similar to our findings, these authors showed that immune system regulation and inflammation pathways include miRNA–mRNA modulatory events. When we compared our data with the report by Bai et al. [26], these were commonly identified in our study, such as the increased expression of miR-155-5p in UC. Interestingly, this miRNA demonstrated a large number of interactions with target genes in our dataset.

Studies have reported changes in miRNA expression in the colonic mucosa of patients with UC, such as over-expression of miR-21 and miR-155, which has been correlated with intestinal inflammation in UC [27]. 

Additionally, miR-155 over-expression has been correlated with intestinal inflammation in 20 patients diagnosed with active UC compared with 16 healthy individuals [28]. Further functional studies assessing the application of miR-155 mimics and inhibitors with a luciferase assay demonstrated that miR-155 specifically regulates forkhead box O 3a (FOXO3a) in the HT29 colorectal adenocarcinoma cell line. The expression of FOXO3a was inversely correlated with miR-155 expression in active UC. The results indicated that miR-155 possesses the capability to bind to and regulate FOXO3a, leading to a reduction in both its gene and protein expression levels. Knockdown of FOXO3a and over-expression of miR-155 also prompted elevated IL-8 levels in TNF-α-treated HT29 cells by inhibiting I kappa B alpha (IκBα). Consequently, it is plausible that miR-155 plays a role in the inflammatory processes within the intestines of individuals with active UC by modulating the downregulation of FOXO3a, which may occur through the activation of Nuclear Factor kappa B (NFκB) [28].

Within the microenvironment of the intestine, extracellular communication serves as the mechanism through which various cell types coordinate biological functions essential for maintaining cellular and tissue physiology. In this intricate microenvironment, the inflammatory response unfolds as a multifaceted process, involving the induction of numerous genes associated with inflammation and immune response activation. This orchestrated response aims to eliminate pathogens and prevent the onset of disease [29]. Notably, miR-155 has emerged as a constituent of the macrophage response to inflammatory mediators, including bacterial lipopolysaccharides (LPS), interferon-β (INF-β), polyriboinosinic polyribocytidylic acid (poly-IC), and TNF-α. Elevated expression of miR-155 has been linked to an augmented release of cytokines during inflammation. Moreover, it has been demonstrated to regulate the intensity of the inflammatory response to microbial stimuli through Toll-like receptors/interleukin-1 (TLR/IL-1) in human dendritic cells [28]. 

Hence, miR-155 emerges as a pivotal miRNA implicated in IBD pathogenesis. The transcription of miR-155 is under the regulation of the activator protein-1 (AP-1) complex [30] and the transcription factor NFκB. miR-155 orchestrates both innate and adaptive immune responses, governing antibody production and cellular cytokine release [31]. Notably, studies have demonstrated that miR-155 facilitates a pro-inflammatory response in a knockout-mouse model (miR-155 −/−) [31], suggesting a potential role of this miRNA in diseases such as IBD.

miR-155 has been unveiled as a regulator of genes within inflammation pathways, including suppressor of cytokine signaling 1 (SOCS1). The silencing of SOCS1 in intestinal myofibroblasts resulted in an augmented release of IL-6 and IL-8. This observation suggests that inflammatory mediators induce the expression of miR-155 in intestinal myofibroblasts of patients with UC. Through the reduction of SOCS1 expression, miR-155 may act as a trigger for the inflammatory phenotype associated with the development of UC [32].

In view of the data presented in this study, we can highlight the importance of exploring and increasingly understanding the role of miRNAs in IBD, especially specific miRNAs for CD and UC, as these can be promising molecular tools in the diagnosis and/or treatment of patients with IBD.

## 5. Conclusions

In conclusion, miRNAs emerge as pivotal effectors in the intricate regulation of inflammation and immune responses, exerting a potent influence on the pathogenesis and progression of various manifestations of IBD. Notably, specific miRNAs present themselves as promising therapeutic targets for IBD, as well as offering a promising avenue for precise modulation of miRNA and target gene expression to effectively manage the inflammatory cascade associated with disease pathogenesis.

## Figures and Tables

**Figure 1 biomedicines-12-00422-f001:**
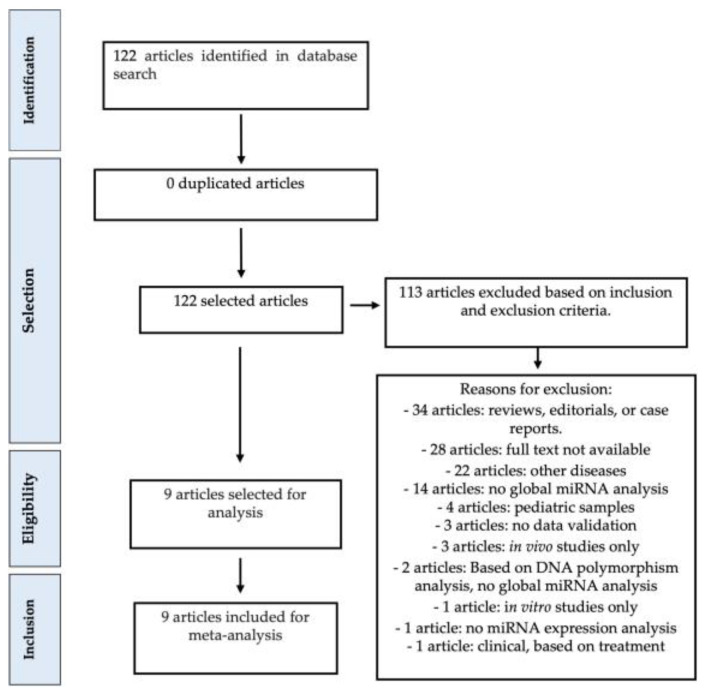
Meta-analysis study design. Quality assessment was performed according to exclusion criteria, as shown above. All articles have been inspected to ensure that miRNA expression data were available for the analyzed samples in each study.

**Figure 2 biomedicines-12-00422-f002:**
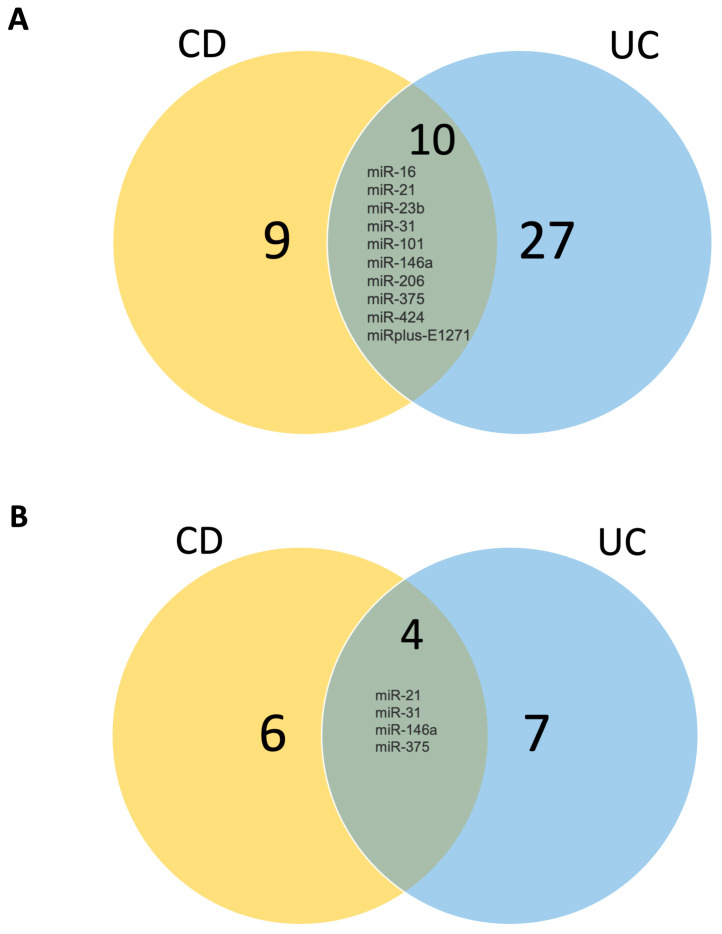
Venn diagram demonstrating (**A**) upregulated miRNAs in CD and UC and (**B**) downregulated miRNAs in CD and UC. UC: ulcerative colitis; CD: Crohn’s disease. Numbers correspond to how many miRNAs are altered in each condition.

**Table 1 biomedicines-12-00422-t001:** Summary of articles included for meta-analysis. Studies are organized in alphabetical order.

Included Studies	Sample	N Screening	Platform Used in Screening	N Validation	Platform Used in Validation
Coskun et al., 2013 [8]	Colonic tissue	Control: 4 CD: 4 UC: 4	Geniom Biochip miRNA array	Control: 20 UC:39	RT-qPCR
Duttagupta et al., 2012 [2]	Blood	Control: 20 UC: 20	Affymetrix Genechip miRNA array	Pooled control samples Pooled UC samples	RT-qPCR
Goten et al., 2014 [10]	Colonic tissue	Control: 10 CD: 5 UC: 17	Affymetrix GeneChip miRNA 2.0 arrays	Control: 10 CD: 5 UC: 17	RT-qPCR
Lin et al., 2014 [11]	Colonic tissue	Control: 18 CD: 9 UC:10	NGS (Illumina)	Control: 29 CD: 26 UC: 36	RT-qPCR
Peck et al., 2015 [12]	Colonic tissue	Control: 14 CD:21	NGS (Illumina)	Control: 15 CD: 20 UC: 6	RT-qPCR
Schaefer et al., 2015 [13]	Blood; Colonic tissue; Saliva	Control: 35 CD: 42 UC: 41	miRCURY LNA microarray (Exiqon)	Pooled control samples Pooled CD samples Pooled UC samples	RT-qPCR
Wu et al., 2008 [3]	Colonic tissue	Control: 15 CD: 5 UC: 30	NCode Multi-Species miRNA microarray Invitrogen	Control: 15 CD: 5 UC: 30	RT-qPCR
Wu et al., 2010 [14]	Colonic tissue	Control: 13 CD: 11	NCode Multi-Species miRNA microarray Invitrogen	Control: 13 CD: 11	RT-qPCR
Wu et al., 2011 [15]	Blood	Control: 13 CD: 19 UC: 23	miRCURY LNA microarray (Exiqon)	Control: 13 CD: 19 UC: 23	RT-qPCR

Deregulated miRNAs considered significant when FC *≥* 2 and *p ≤* 0.05 vs. control group. NGS: next-generation sequencing; UC: ulcerative colitis; CD: Crohn’s disease; IC: infectious colitis; Control: healthy subjects.

**Table 2 biomedicines-12-00422-t002:** Deregulated miRNAs in plasma and tissue biopsies of patients with Crohn’s disease.

Upregulated	Reference	Downregulated	Reference
miR-16	[14]	miR-19	[14]
miR-21 (Colonic Tissue)	[13,14]	miR-21 (Blood)	[13]
miR-23b	[14]	miR-31 (Blood)	[13]
miR-31 (Colonic Tissue)	[11,12,13]	miR-146a (Blood)	[13]
miR-101	[13]	miR-149-3p	[15]
miR-106	[14]	miR-149-5p	[12]
miR-146a (Colonic Tissue)	[11,13]	miR-155	[13]
miR-191	[14]	miR-375 (Colonic Tissue)	[13]
miR-199a-5p	[15]	miR-629	[14]
miR-206	[11]	miRplus-F1065	[15]
miR-215	[12]		
MiR-223	[14]		
miR-340-3p	[15]		
miR-362-3p	[15]		
miR-375 (Blood)	[13]		
miR-424	[11]		
miR-532-3p	[15]		
miR-594	[15]		
miRplus-E1271	[15]		

Considered significant when FC ≥ 2 and *p* ≤ 0.05.

**Table 3 biomedicines-12-00422-t003:** Deregulated miRNAs in plasma and tissue biopsies of patients with ulcerative colitis.

Upregulated	Reference	Downregulated	Reference
let 7e* Let-7f miR-101 miR-103-2*-5p mir-125b-1* miR-126-3p miR-126-5p miR-142-3p miR-151-5p miR-155-5p miR-16-5p miR-195-5p miR-199a-5p miR-19a miR-206 miR-20b miR-223 miR-23a-3p miR-23b-3p miR-24-3p miR-28-5p miR-29a-3p miR-340-3p miR-362-3p miR-375 (Colonic tissue, blood) miR-378 miR-424 miR-494 miR-532-3p miR-650 miR-874 miR-98 miR-142-5p (Blood) miR-146a (Colonic tissue) miR-21-5p (Colonic tissue, saliva) miR-31-5p (Colonic tissue, saliva)	[8] [3] [13] [15] [8] [3] [3] [13] [15] [10] [3] [3] [15] [13] [11] [8] [13] [3] [3] [3] [15] [3] [15] [15] [10,13] [2] [11] [13] [15] [10] [2] [8] [13] [10,11,13] [3,10,13] [10,11,13]	miR-21-5p (Blood) miR-31-5p (Blood) miR-142-5p (Saliva) miR-146a (Blood) miR-192 -5p miR-196b-3p miR-196b-5p miR-200c-3p miR-375 (Colon) miR-422b miR-505-5p	[3,10,13] [10,11,13] [13] [10,11,13] [3] [10] [10] [10] [3] [3] [15]

Considered significant when FC ≥ 2 and *p* ≤ 0.05.

## Data Availability

The dataset analyzed during the current study is available from the corresponding author Rogerio Saad-Hossne on reasonable request.

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
