# Peer review of "microRNA–mRNA Networks Linked to Inflammation and Immune System Regulation in Inflammatory Bowel Disease"

_biomedicines, 2024, doi:10.3390/biomedicines12020422_

Round 1

Reviewer 1 Report

Comments and Suggestions for Authors

In this study, Carina et al. aimed to investigate the miRNA-mRNA network in Inflammatory bowel Disease. They performed a meta-analysis on Crohn´s Disease and Ulcerative Colitis and identify several miRNAs. They just search the NCBI database to get 9 studies and performed the miRNA targeted gene prediction. Compared to similar publication in MDPI journal, I think their works are not enough to publish in our journal.

other comments:

1. Line 29 and other line: “09 studies”, should be “9 studies”

2. please re-arrange format of Table 1 and Table 2, at least list the p-value and FC value

3. Figure 2 and Figure 3 can be combined into one figure

Author Response

Response to Reviewer 1

Comments and suggestions for authors
R1 Comment 1: Line 29 and other line:“09 studies”, should be “9 studies”

Answer: We thank the reviewer for the suggested correction, which has been done to improve readability of our manuscript.

R1 Comment 2: Please re-arrange format of Table 1 and Table 2, at least list the p-value and FC value.

Answer: Tables 1 and 2 were revised to add p-values and FC values, as recommended. R1 Comment 3: Figure 2 and Figure 3 can be combined into one figure.

Answer: We have combined the two Figures, as suggested by the reviewer.

Reviewer 2 Report

Comments and Suggestions for Authors

Estimated Authors,

I've read with great interest the present study from de Sibia et al that, through a rigorous meta-analytical approach, tentatively identified 11 over-expressed  and 5 under-expressed miRNA associated with both UC and CD, and a series of miRNA more precisely associated with either UC or CD.

The study is significant for its content and its appropriate design. In fact, I've detected no significant flaws, only a series of minor amendments are, by my opinion, recommended.

1) Table 1: reporting a column entitled "p value" may be quite misleading; please explain which analysis does the "p value" refer to.

2) regarding Table 2 and Table 3, could you provide (also as Annex Table) an amended version reporting which studies from Table 1 are describing the assessed sequences? (e.g. miR-140-3p [xx, xy, zz], etc).

Thank you again: from my point of view, a well written and interesting study.

Author Response

Response to Reviewer 2

Comments and suggestions for authors

R2 Comment 1: Table 1: reporting a column entitled "p value" may be quite misleading; please explain which analysis does the "p value" refer to.

Answer: MicroRNAs shown in Table 1 and Table 2 were considered significant when FC≥2 and p≤0.05 compared with control group (i.e healthy people). In order to address the reviewer ́s comment, we added the information regarding the p-value analysis in the table heading.

R2 Comment 2: Regarding Table 2 and Table 3, could you provide (also as Annex Table) an amended version reporting which studies from Table 1 are describing the assessed sequences? (e.g. miR-140-3p [xx, xy, zz], etc).

Answer: We added this information, as requested. Please refer to Tables 1 and 2 in the manuscript. We would like to take the opportunity to justify the changes made in Tables 2 and 3, as when reviewing each miRNA thoroughly, we observed that some miRNAs presented in Tables 2 and 3 were not validated and therefore we had to exclude them. We would like to apologize for this error, which has now been duly corrected.

Reviewer 3 Report

Comments and Suggestions for Authors

The review article “Identification of microRNA-mRNA networks associated with inflammation and immune system regulation in Inflammatory  Bowel Disease” is an interesting article, I have the following comments/suggestions,

The title is appealing and clear.

The abstract is very well written and gives a good idea of the work done.

The introduction section is well-written and provides sufficient background on the topic and the gaps and objectives are clear.

The method section includes all the details for the performed literature search and data extraction,  but I am not seeing the quality check for the final included studies as it will be more interesting for the readers to see the quality assessment of the final included articles.

The results are well explained, and tables are well placed and figures are informative.

The discussion and conclusion are in line with the presented results.   

Author Response

Response to Reviewer 3

Comments and suggestions for authors

R3 Comment 1: The review article “Identification of microRNA-mRNA networks associated with inflammation and immune system regulation in Inflammatory Bowel Disease” is an interesting article, I have the following comments/suggestions: The title is appealing and clear. The abstract is very well written and gives a good idea of the work done. The introduction section is well-written and provides sufficient background on the topic and the gaps and objectives are clear. The method section includes all the details for the performed literature search and data extraction, but I am not seeing the quality check for the final included studies as it will be more interesting for the readers to see the quality assessment of the final included articles. The results are well explained, and tables are well placed and figures are informative. The discussion and conclusion are in line with the presented results.

Answer: We thank the reviewer for the kind remarks. In order to address the reviewer comment regarding quality check for the final included studies, we would like to clarify that quality assessment has been performed when we applied the “exclusion criteria” to 122 selected studies, which resulted in the exclusion of 113 studies due to the following issues: 34 articles were reviews, editorials, or case reports without having generated original data; 28 articles did not have full text available, which did not allow us to verify the study findings; 22 articles were related to investigation of other diseases; 14 articles did not perform global microRNA analysis; 4 articles referred to study of paediatric patients; 3 articles did not include data validation; 3 articles were based solely on in vivo studies; 2 articles were based on DNA polymorphism analysis only; 1 article performed in vitro studies only without using clinical samples; 1 article did not include any type of microRNA expression analysis and 1 article was a clinical study based on treatment. In order to make this point clear to readers, we have included a statement in Figure 1 legend, as follows: “Quality assessment was performed according to exclusion criteria, as shown above. All articles have been inspect-ed to ensure that miRNA expression data was available for the analyzed samples in each study.”

Please refer to Figure 1 in the manuscript.

Reviewer 4 Report

Comments and Suggestions for Authors

The authors examined the involvement of the expression of many microRNA genes, mainly focusing on the identification of deregulated miRNAs, defined as miRNAs with a fold change in expression ≥2 (underexpression or overexpression) compared to the control group. They selected articles presenting changes for meta-analysis
in global miRNA expression patterns in plasma and intestinal disease tissues in individuals diagnosed with Crohn's disease (CD) and/or ulcerative colitis (UC). A layered approach based on disease classification, we successfully divided miRNA expression data into two distinct categories: those related to CD and those related to
I ate with UC. This stratified analysis facilitated the identification of miRNAs showing disease-specific deregulation as well as those showing common deregulation across both conditions. Based on this analysis, 32 deregulated miRNAs were identified in CD patients, including 17 overexpressed and 15 underexpressed miRNAs. A more extensive miRNA deregulation profile, with 93 miRNAs showing altered expression, exceeding the number observed in CD patients. Visually, representations of the overlapping and unique sets of overexpression and underexpression of miRNAs in these diseases can be observed and are shown in Figures 2 and 3, respectively. Very general conclusions, but no other conclusions can be drawn based on the analysis. MiRNAs appear to be key effectors in the complex regulation of inflammation and immune responses, and exert a strong influence on the pathogenesis and progression of various IBD symptoms.

Comments on the Quality of English Language

none

Author Response

Response to Reviewer 4

Comments and suggestions for authors

R4 Comment 1:

The authors examined the involvement of the expression of many microRNA genes, mainly focusing on the identification of deregulated miRNAs, defined as miRNAs with a fold change in expression ≥2 (underexpression or overexpression) compared to the control group. They selected articles presenting changes for meta-analysis in global miRNA expression patterns in plasma and intestinal disease tissues in

individuals diagnosed with Crohn's disease (CD) and/or ulcerative colitis (UC). A layered approach based on disease classification, we successfully divided miRNA expression data into two distinct categories: those related to CD and those related to I ate with UC. This stratified analysis facilitated the identification of miRNAs showing disease-specific deregulation as well as those showing common deregulation across both conditions. Based on this analysis, 32 deregulated miRNAs were identified in CD patients, including 17 overexpressed and 15 underexpressed miRNAs. A more extensive miRNA deregulation profile, with 93 miRNAs showing altered expression, exceeding the number observed in CD patients. Visually, representations of the overlapping and unique sets of overexpression and underexpression of miRNAs in these diseases can be observed and are shown in Figures 2 and 3, respectively. Very general conclusions, but no other conclusions can be drawn based on the analysis. MiRNAs appear to be key effectors in the complex regulation of inflammation and immune responses, and exert a strong influence on the pathogenesis and progression of various IBD symptoms.

Answer: We express our gratitude to the reviewer for their insightful remarks and key points. Following the reviewer ́s feedback, we have revised the manuscript text and added a conclusive statement at the end of the Discussion section to augment its comprehensiveness.

Round 2

Reviewer 1 Report

Comments and Suggestions for Authors

The authors have well replied my comments and revised many parts of the manuscript, thus, it can be accepted for publication.